# Assessment of Antioxidant and Antimicrobial Properties of Selected Greek Propolis Samples (North East Aegean Region Islands)

**DOI:** 10.3390/molecules27238198

**Published:** 2022-11-24

**Authors:** Elisavet Pyrgioti, Konstantia Graikou, Antigoni Cheilari, Ioanna Chinou

**Affiliations:** Lab of Pharmacognosy and Chemistry of Natural Products, Faculty of Pharmacy, National and Kapodistrian University of Athens, 15771 Athens, Greece

**Keywords:** northeast Greek Aegean islands, propolis, GC-MS, antioxidant activity, antimicrobial activity

## Abstract

Propolis is a bee-produced substance rich in bioactive compounds, which has been utilized widely in folk medicine, in food supplement and cosmetology areas because of its biological properties, (antibacterial, antiviral, antioxidant, anti-inflammatory, etc.). The subject of this study is associated with the chemical analysis and the biological evaluation of 16 propolis samples from the northeast Aegean region Greek islands, a well-recognized geographic area and the homeland of rich flora as a crossroads between Europe and Asia. Our study resulted in the detection of a significant percentage of diterpenes by gas chromatography–mass spectrometry (GC-MS), while flavonoids were identified in low percentages among studied samples. Furthermore, the DPPH assay highlighted that eight of the samples (Lesvos and Lemnos origin) demonstrated a promising antioxidant profile, further verified by their total phenolic content (TPC). Additionally, the propolis samples most rich in diterpenes showed significant antibacterial and fungicidal properties against human pathogenic microorganisms, proving them to be a very interesting and promising crude material for further applications, concluding that floral diversity is the most responsible for the bioactivity of the propolis samples.

## 1. Introduction

Propolis is a balsamic and resinous multicomponent natural substance produced by honeybees (*Apis mellifera*) using different plant exudates. It is characterized by a variable chemical composition as it depends on the plant source of each region and the collection period [1,2]. Following the literature, propolis consists of more than 180 different chemical compounds [3,4,5] and it is well-known for its health properties, according to which propolis has been exploited by humans in medicine and cosmetology since antiquity [6]. Nowadays, propolis is widely chemically and pharmacologically studied for its biological effects, which are mostly antioxidant, antibacterial, antiviral, immunomodulatory and anti-inflammatory properties, while is widely used as a wound-healing agent [7,8,9]. Furthermore, propolis is among the studied immunomodulating agents potentially active against COVID-19 disease as a supplementary treatment because of its positive feedback toward the reduction in the length of hospitalization [10,11].

The Greek islands of the northeast Aegean region (NEAR) are a group of nine inhabited islands (Lemnos, Agios Efstratios, Lesvos, Chios, Psara, Oinousses, Samos, Ikaria and Fourni), which are located in the northern part of the Aegean Archipelago (Figure 1), thus constituting a crossroads between Europe and Asia. The area covers an area of 3835 km^2^ and has about 2500 vascular plant species. The territory of the islands is 33% mountainous, 35% hilly and 32% flat. The surface and the maximum height range from 40 km^2^ for Psara and Fourni, to 1633 km^2^ for Lesvos and from 0 to 1433 m high (Kerketea mt, Samos), respectively. The NEAR islands offer a unique ecosystem with several endemic and endangered species, depending also on the fact that the Aegean Sea hosts an archipelago placed at the summit of Europe, Asia and Africa [12,13]. Generally, Greece is one of the most plant-species-rich European countries and, as biodiversity hotspots are usually located near mountainous areas and islands, some Aegean islands (Lesvos and Samos) are important Greek endemic hotspots [14]. It is noteworthy that among the NEAR islands, Chios is famous all over the world for the emblematic product of mastic resin, as is the island of Lemnos for the high level of the ecological value that is due to the existence of a variety of different vegetation. Furthermore, both the islands of Chios and Samos belong to the category ‘Flora of the East Aegean Islands’ used in the World Geographical Scheme for Recording Plant Distributions, while the island of Ikaria is considered a “blue-zone”, a designation for places of exceptional longevity, one among only five distinguished existing worldwide.

The NEAR islands can be grouped into two different phytogeographical zones (the zone of the northern Aegean with Lemnos and Agios Efstratios and the zone of the eastern Aegean with Lesvos, Chios, Psara, Oinousses, Samos, Ikaria and Fourni) among the 13 that exist in Greece [12,15].

In the framework of our studies on selected natural products from NEAR [16,17], as well as on propolis from Greece and worldwide [1,3], we present herein the chemical analyses of 16 samples from this area and the evaluation of their biological effects, to our knowledge for the first time highlighting the big problem on the further exploitation of propolis because of its varied composition that depends on the flora of the geographic area, annual weather conditions, potential environmental pollution, apicultural followed practices and inclusion of polluting waxes/other external materials, among several others.

## 2. Results

### 2.1. Propolis Composition

#### 2.1.1. Gas Chromatography–Mass Spectrometry (GC-MS) Analysis

The chemical composition of the studied propolis samples (70% ethanolic extracts) was investigated by GC-MS after silylation (Appendix A). The main chemical classes of the identified compounds are listed in Table 1. The essential characteristic is the existence of diterpenes in all samples and the low amounts or in some cases the absence of flavonoids, which categorize them in the Mediterranean type of propolis [3].

#### 2.1.2. Isolation of Chemical Constituents

In our continuous and systematic studies on propolis from different countries all over the world [1,3,18,19,20], we use as internal standards the isolated metabolites from the different studied propolis samples. In this research, further analysis was performed on the sample POi, one of the most diterpene-rich samples from an extremely rarely studied area. Compounds of diterpene structure (Appendix A) such as totarol, manoyl-oxide, ferruginol, epitorulosol, 13-epitorreferol, agathadiol, manool, copalol, 14,15-dinor-13-oxo-8(17)labden-19-oic acid, pimaric acid, imbricataloic acid and 13-epi-cupressic acid were isolated and determined through GC-MS.

### 2.2. Antimicrobial Activity

All the ethanolic extracts of the propolis samples were evaluated for their antimicrobial activity by the diffusion and dilution method against eight Gram-negative and -positive bacterial strains and three human-pathogenic fungi. The results of these tests (Table 2) showed significant and promising antibacterial activity for the samples PCh1 and 2, as well as for PFo and PSa. Moreover, an association was found between the levels of diterpenes of all samples and the antimicrobial activity (Figure 2). Spearman’s rank correlation coefficients were −0.88 to −0.64, indicating a strong correlation between the concentration of diterpenes and antimicrobial activity (*p* < 0.005). In Figure 2, the red and blue colors represent negative and positive Spearman’s rank correlation coefficients between compound concentration levels and activity, respectively. In the case of antimicrobial activity, the negative correlation means that the antimicrobial activity is higher when diterpene levels are higher.

### 2.3. Total Phenolic Content (TPC) and DPPH Radical Scavenging Activity

TPC of propolis extracts (Table 3) was determined by the Folin–Ciocalteu method [21]. The results of the assay showed that samples PLs, PLm and PIk showed the highest phenolic content. The majority of the remaining samples showed a moderate to low phenolic profile, which can be explained by their chemical composition. The DPPH radical scavenging activity of the ethanolic extracts was found to exhibit a large range of inhibition, while the samples from Lemnos (PLm1 and 2) showed the highest inhibition (more than 90% inhibition at 200 μg/mL) among all. The samples of Lesvos (PLs1–6), with the exception of PLs7, show moderate inhibition (55–72% inhibition at 200 μg/mL), while the rest of the samples are characterized as inactive (4–40% inhibition at 200 μg/mL). Furthermore, sugars seem to correlate (rs 0.52–0.58) positively (blue color in Figure 2) with TPC and DPPH inhibition, while diterpenes were inversely (red color in Figure 2) correlated, as expected (*p* < 0.01).

## 3. Discussion

Diterpenes are identified in all NEAR propolis samples, in some cases with significant amounts ranging up to 58%, while, in parallel, minor percentages (up to 10% only) or absence of flavonoids and chalcones were detected. It is noteworthy that the highest percentages of diterpenes were detected in the samples from the islands of Samos (58.02%), Chios (56.67% and 50.23%), Oinousses (53.81%), Fourni (50.65%) and Ikaria (30.82%), where all islands are located in the southeastern part of the NEAR. Furthermore, these islands show a geographical relevance and, consequently, a similar profile of vegetation and local flora.

The rich diterpenic profile (with isocupressic acid, pimaric acid, communic acid, isoagatholal, agathadiol and totarol as major components) of these samples could categorize them as Mediterranean-type propolis [1,3], which is well-known for its high antimicrobial properties and weaker antioxidative ones.

The low concentration or the lack of flavonoids could be explained by the non-spread of poplar trees (genus *Populus*) as a feeding source for bees in the respective collection areas. At the same time, according to the literature data, such diterpenes as the ones found in our samples are derived mainly from Pinaceae and Cupressaceae plant families, which are widespread in Greece [1,3], while *Pinus* and *Juniperus* are widespread also in the NEAR (e.g., Chios island) [22]. Botanical research on the island of Ikaria has shown that it has strong phytogeographical links with the islands of the eastern Aegean and Anatolia, a fact that can be attributed to the relatively recent land connection between these areas during the Pleistocene [23]. The appearance of a rich diterpenic profile in the sample from Oinousses is probably justified by the appearance of the perennial species *Pinus brutia*, which is observed all around the port village of the main island [24]. Regarding the samples from Lesvos, the presence of diterpenes can be partially justified by the flora of the island, as some areas between the two bays to the north and southeast are covered by a pine forest (*Pinus brutia*) [25].

Furthermore, the presence of diterpenes in the samples (concentration levels) was highly correlated (Spearman r) with their antimicrobial properties and anti-correlated with TPC and DPPH radical scavenging activity. The latter, as expected, were associated with the presence of sugars in the samples.

It deserves to be mentioned that an unexpected increased percentage of glycerol, a substance naturally identified in small percentages, was detected in the majority of the studied propolis, which can be explained by the beekeeping technique with extensive use of oxalic acid strips (impregnated with glycerin) for the fight against *Varroa* mites. It is noted that neither honey nor wax are affected by this technique [26], but it is important in the case of propolis because of the impact on the propolis composition quality [20]. As those high percentages of glycerin do not come from nature, the percentages have been eliminated from the analytical results.

Based on antimicrobial assays, it is observed that some samples from the NEAR revealed strong antibacterial and antifungal activity, which is completely connected with the detected high percentage of diterpenes in the studied samples and could be attributed to them [3,27]. Samples from Chios (PCh1-2), Fourni (PFo), Samos (PSa), Oinousses (POi) and Ikaria (PIk) seem to be the most active overall, displaying the lowest MIC values in all tests, fully consistent in their chemical composition, as they are the ones with the highest diterpene percentages.

The DPPH radical scavenging activity of the extracts at the concentration of 200 μg/mL showed an interesting inhibition (more than 55%) for Lemnos and Lesvos propolis samples and a low inhibition (less than 40%) for the rest of studied samples.

It seems that the studied propolis, which belong to the Mediterranean type as they display significant amounts of diterpenes and a relatively low quantity of phenolic acids and their esters, are divided in two groups according to their studied activity: the northern part of NEAR (Lemnos and Lesvos) show significant antioxidant activity, while the southern part (Chios, Fourni, Samos, Oinousses and Ikaria) show strong antimicrobial activity.

## 4. Materials and Methods

### 4.1. Samples

Sixteen propolis samples were provided by local beekeepers from the NEAR islands (Table 4) and kept in a refrigerator (4 °C) till the analyses.

### 4.2. Extraction and Sample Derivatization

Propolis samples (10 g) were extracted three times with 70% ethanol (1:10, *w:v*) by maceration at room temperature for 24 h, followed by filtration of the resulting suspension at room temperature using a paper filter and in vacuum evaporation of the solvent to dryness on a rotary evaporator. About 5 mg of each residue was mixed with 40 µL of dry pyridine and 50 µL of BSTFA (bis(trimethylsilyl) trifluoracetamide) and heated at 80 °C for 20 min before GC-MS analysis.

### 4.3. GC-MS Analysis

The chemical analysis was performed by the technique of gas chromatography coupled with mass spectrometry (gas chromatography–mass spectrometry, GC-MS). The analysis was performed on an Agilent 7820A gas chromatograph, connected to an Agilent 5977B mass spectrometer system (Agilent Technologies, Santa Clara, CA, USA) based on electron impact (EI) and 70 eV ionization energy. The gas chromatograph was also equipped with a split/splitless injector and a capillary column HP5MS 30 m, internal diameter 0.25 mm and membrane thickness 0.25 μm. The temperature was programmed from 100 to 300 °C at a rate of 5 °C/min. The carrier gas was He at a flow rate of 0.7 mL/min, injection volume of 2 μL, split ratio of 1:10 and injector temperature of 280 °C. The identification was accomplished using Wiley mass spectral databases (and database created by our research team). The components of propolis extract were determined by considering their areas as percentages of the total ion current.

### 4.4. Isolation of Compounds

The POi ethanol extract was submitted to column chromatography. Therefore, 763.5 mg of extract was subjected to vacuum liquid chromatography with a stationary gel phase silica 60 H (25.0 g) and a mobile phase with solvents cyclohexane, dichloromethane and ethyl acetate in increasing polarities. The 125 fractions derived from the column were evaluated through thin-layer chromatography (TLC-aluminum sheets coated with silica gel 60 F254 Merck) and were visualized under UV light (254 nm and 366 nm) after spraying with vanillin in sulfuric acid, followed by heating at 100 °C. The fractions were combined into groups of similar chemical profile according to TLC examination. Several metabolites were isolated and determined as totarol (3.3 mg), manoyl-oxide (2.5 mg), ferruginol (5.9 mg), epitorulosol (1.4 mg), 13-epitorreferol (1.1 mg), agathadiol (1.0 mg), manool (1.3 mg), copalol (1.7 mg), 14,15-dinor-13-oxo-8(17)labden-19-oic acid (4.5 mg), pimaric acid (2.8 mg), imbricataloic acid (3.1 mg) and 13-epi-cupressic acid (3.8 mg), all of the diterpene chemical type (Appendix A). GC-MS was used as the identification method and the data were compared to the bibliographic data [1,3,4] and the internal databases.

### 4.5. Antimicrobial Bioassay

All extracts were investigated for their antimicrobial activity against the two Gram-positive bacteria *Staphylococcus aureus* (ATCC 25923) and *S. epidermidis* (ATCC 12228), the two Gram-positive oral bacteria *S. mutans* and *S. viridians*, the four Gram-negative bacteria *Escherichia coli* (ATCC 25922), *Enterobacter cloacae* (ATCC 13047), *Klebsiella pneumoniae* (ATCC 13883) and *Pseudomonas aeruginosa* (ATCC 227853) and the three pathogen fungi *Candida albicans* (ATCC 10231), *C. tropicalis* (ATCC 13801) and *C. glabrata* (ATCC 28838). All studied samples dissolved in dimethyl sulfoxide (DMSO) were screened for in vitro antibacterial and antifungal activities in Mueller–Hinton or Sabouraud broths, as previously described [20].

First, the assays were carried out by the disc diffusion method measuring the zone of inhibitions. For each experiment, control disks with pure solvent were used as a blind control. Petri dishes had been previously inoculated with the tested microorganisms to give a final cell concentration of 107 cells/mL. Of the above solutions, 10 µL were required to wet (impregnate) the test paper discs. The incubation conditions used in the experiments were 24 h at a temperature of 37 °C. The growth conditions and the sterility of the medium of each strain were controlled and then the plates were incubated. The results were reported as the diameter of the zone of inhibition around each disk (in mm).

The minimal inhibitory concentrations (MIC) of the tested extracts were evaluated by the broth micro-dilution method. The sterile 96-well polystyrene microtitrate plates were prepared by dispensing 100 µL of the appropriate dilution of the tested extracts in a broth medium, per well, to obtain the final concentrations of the tested extracts that ranged from 0.50 to 10 mg/mL. The inoculums that were prepared with fresh microbial cultures in sterile 0.85% NaCl, to match the turbidity of the 0.5 McFarland standard, were added to the wells to obtain a final density of 1.5 × 106 CFU/mL for bacteria and 5 × 104 CFU/mL for yeasts (CFU: colony forming units). After incubation (37 °C for 24 h), the MICs were assessed visually for the lowest concentration of the extracts, showing the complete growth inhibition of the reference microbial strains. An appropriate DMSO control (at a final concentration of 10%), a positive control (containing the inoculum without the tested samples), and the negative control (containing the tested derivatives without the inoculum) were included on each microplate.

Standard antibiotic netilmicin (at concentrations 4–88 μg/mL) was used to control the sensitivity of the tested bacteria and saguinarine for oral bacteria, whilst 5-flucytocine and itraconazole (at concentrations of 0.5–25 μg/mL), as well as amphotericin B, were used as controls against the tested fungi (Sanofi, Diagnostics Pasteur at concentrations of 30, 15 and 10 μg/mL). For each experiment, any pure solvent used was also applied as blind control. The experiments in all cases were repeated three times and the results were expressed as mean values.

### 4.6. Total Phenolic Content (TPC)

The total phenolic content of the samples was determined by the Folin–Ciocalteu method [28]. In a 96-well plate, 25 μL of propolis extracts of different concentrations (4, 2 and 1 mg/mL) or standard solutions of gallic acid (2.5, 5, 10, 12.5, 20, 25, 40, 50, 80 and 100 g/mL), both diluted in DMSO, were added to 125 μL of a Folin–Ciocalteu solution (10%), followed by the addition of 100 μL of 7.5% sodium carbonate [17]. The plate was incubated for 30 min in darkness at room temperature. The absorbance at 765 nm was measured using a TECAN Infinite m200 PRO multimode reader (Tecan Group, Männedorf, Switzerland). All measurements were performed in triplicate, with the mean values plotted on a gallic acid calibration curve, and the total phenolic content was expressed as mg equivalent to gallic acid (GAE) per gram of dry extract.

### 4.7. DPPH (2,2-DiPhenyl-1-PicrylHydrazyl) Assay

To determine the antioxidant activity, different propolis extracts (4, 2 and 1 mg/mL) were prepared using DMSO as a solvent. In a 96-well plate, 10 μL of each sample were mixed with 190 μL of DDPH solution (12.4 mg/100 mL in ethanol) and then incubated at room temperature for 30 min, strictly in darkness. The absorbance was measured at 517 nm. All measurements were performed in triplicate and gallic acid was used as positive control [21]. The % inhibition of the DPPH radical for each dilution was calculated using the following formula: %Inhibition = {[1 − (A − AB)]/AT} × 100, where A is the absorbance of the sample, AT the absorbance of control and AB the absorbance of the sample without the DPPH radical.

### 4.8. Statistical Analysis

Correlation of the propolis chemical content with antimicrobial activity, TPC and DPPH scavenging capacity was performed in GraphPad Prism 8 (GraphPad Software, San Diego, CA, USA). The correlation coefficient (Spearman r) was computed for each pair of variables because of the non-normal distribution of samples; the confidence interval was 95% for *p* values and a heatmap of the correlation matrix was generated. Only *p* < 0.05 correlations were taken into consideration.

## 5. Conclusions

As propolis is an important health-promoting agent, the overall aim of this study was to investigate, chemically and biologically, different propolis samples from the islands of the northeast Aegean region for the first time. All propolis samples appeared to belong to the Mediterranean-type profile, with characteristic diterpene composition, while according to their bioactivity they are divided in two groups: propolis from the northern part of NEAR (Lemnos and Lesvos) showing significant antioxidant activity and propolis from the southern part (Chios, Fourni, Samos, Oinousses and Ikaria) showing strong antimicrobial activity.

Promising results on NEAR propolis call for further exploitation, mostly based on their antimicrobial properties, reinforcing the view, in agreement with very recent overview article [29], that this natural product from such an ecosystem/geographical area (Mediterranean type) should be studied in greater detail because of its high scientific interest. Furthermore, toward the potential marketing of propolis as a medicine, health supplement and/or cosmetic, it would be essential to standardize its chemical quality, based on a strategy that includes botanical, geographical and/or biological activity multi-marker(s). Moreover, due to propolis’s broad spectrum of commercial applications, side effects such as allergic reactions demand further in vitro and in vivo experiments, as well as clinical studies, to strengthen the safe use of this high-value natural product.

## Figures and Tables

**Figure 1 molecules-27-08198-f001:**
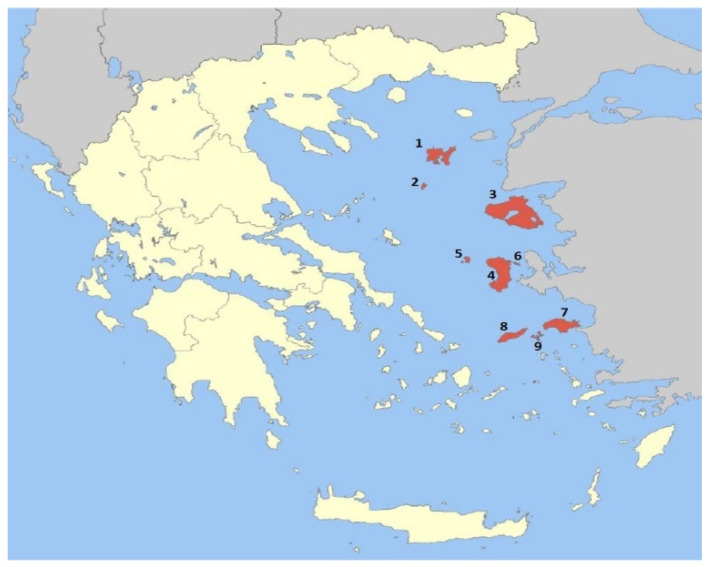
The northeast Aegean islands: 1. Lemnos, 2. Agios Efstratios, 3. Lesvos, 4. Chios, 5. Psara, 6. Oinousses, 7. Samos, 8. Ikaria, 9. Fourni. https://en.wikipedia.org/wiki/North_Aegean#/media/File:Vorio_Egeo_in_Greece.svg (17 October 2022).

**Figure 2 molecules-27-08198-f002:**
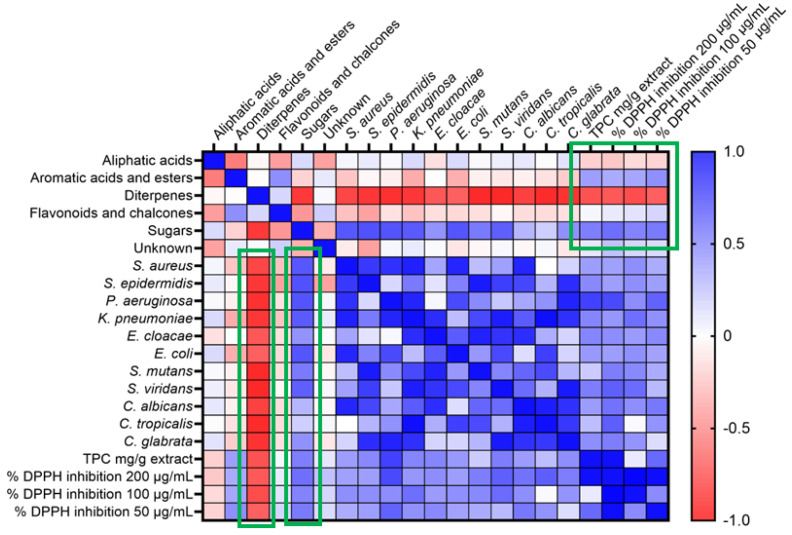
Propolis samples’ chemical content correlates with antimicrobial activity, TPC and DPPH inhibition activity. Red and blue colors represent negative and positive Spearman’s rank correlation coefficients, respectively. Only *p* < 0.05 correlations (green highlight) were taken into consideration.

**Table 1 molecules-27-08198-t001:** Chemical categories of compounds from NEAR propolis.

%	Aliphatic Acids	Phenolic Acids and Esters	Diterpenes	Flavonoids and Chalcones	Sugars
PLs1	7.28	4.33	11.96	0.48	71.50
PLs2	3.96	0.89	13.68	0.21	72.69
PLs3	14.19	0.88	4.76	-	78.23
PLs4	12.22	4.30	7.51	0.65	67.40
PLs5	4.95	-	5.17	-	68.62
PLs6	6.22	4.71	13.80	-	70.77
PLs7	4.03	8.79	19.64	8.85	48.50
PLm1	5.48	14.95	3.29	-	71.05
PLm2	3.19	27.59	1.94	2.88	45.31
PCh1	18.13	-	56.35	-	16.20
PCh2	5.92	1.97	50.23	10.60	11.57
PFo	15.38	4.50	47.36	-	22.07
PPs	6.75	0.19	0.70	-	89.83
PIk	3.49	6.52	31.43	-	53.63
POi	5.16	-	53.81	-	39.05
PSa	1.36	13.18	58.02	6.63	8.72

**Table 2 molecules-27-08198-t002:** Antimicrobial activities (zones of inhibition in mm/and MIC mg/mL, *n* = 3).

	*S. aureus*	*S. epidermidis*	*P. aeruginosa*	*K. pneumoniae*	*E. cloacae*	*E. coli*	*S. mutans*	*S. viridans*	*C. albicans*	*C. tropicalis*	*C. glabrata*
PLs1	14/0.79	14/0.81	12/1.79	13/1.67	11/0.92	13/1.69	14/0.79	15/0.96	10/1.90	10/1.85	11/1.83
PLs2	13/0.88	14/0.85	11/1.92	12/1.79	11/0.97	12/1.73	13/0.88	14/1.00	10/1.95	10/1.90	10/1.94
PLs3	12/1.13	12/0.99	10/1.97	10/1.98	10/1.84	11/1.87	12/1.17	12/1.12	9/2.24	10/1.97	10/2.14
PLs4	12/1.22	12/1.20	10/1.99	10/1.93	10/1.95	10/2.04	12/1.64	12/1.55	9/2.42	10/2.05	10/1.99
PLs5	12/1.17	13/0.88	10/1.89	10/1.95	10/1.89	11/1.95	12/1.03	13/0.97	9/2.20	10/2.14	11/1.90
PLs6	12/1.00	12/0.99	11/1.88	10/1.82	11/1.49	10/1.94	12/0.99	13/0.72	9/2.29	10/1.95	10/1.91
PLs7	13/0.92	14/0.84	12/1.67	12/1.55	12/1.32	12/1.58	13/0.84	14/0.63	10/2.00	11/1.74	12/1.56
PLm1	12/0.99	12/1.15	10/2.02	11/1.88	10/1.93	12/0.89	12/1.20	12/1.18	9/2.02	10/1.95	11/1.87
PLm2	12/0.97	12/0.95	10/1.97	11/1.92	11/1.95	12/0.94	12/1.31	12/1.25	9/2.15	10/1.99	11/1.92
PCh1	22/0.12	23/0.09	16/1.14	17/1.00	15/0.72	19/0.67	18/0.52	18/0.39	15/0.94	17/0.27	17/0.22
PCh2	22/0.14	22/0.11	15/1.15	16/1.10	15/0.75	18/0.71	17/0.44	17/0.42	14/1.00	17/0.38	16/0.28
Pfo	21/0.17	22/0.15	15/1.00	16/0.97	15/0.69	18/0.48	17/0.40	18/0.31	14/0.98	17/0.35	18/0.20
PPs	11/1.16	11/1.24	10/1.88	10/1.92	12/1.18	11/1.80	11/1.28	12/1.12	9/2.11	11/1.95	11/1.98
Pik	15/0.88	16/0.90	15/1.15	15/0.98	14/1.17	15/0.84	17/0.61	18/0.39	12/1.23	14/0.88	14/0.77
Poi	15/0.92	17/0.85	12/1.75	12/1.80	12/1.87	14/0.94	16/0.81	17/0.65	10/1.57	12/1.44	13/1.30
Psa	18/0.02	19/0.01	16/0.06	16/0.05	16/0.05	15/0.08	18/0.03	19/0.01	16/0.05	16/0.05	15/0.08

**Table 3 molecules-27-08198-t003:** Total phenolic content and antioxidant activity as % DPPH inhibition of the studied samples.

Samples	TPC mg GAE/g Extract	% DPPH Inhibition		
		200 μg/mL	100 μg/mL	50 μg/mL
PLs1	109.66 ± 0.64	72.57 ± 1.34	39.12 ±0.78	21.66 ± 0.57
PLs2	101.78 ± 0.64	65.64 ± 0.55	35.47 ±1.39	18.51 ± 0.36
PLs3	105.48 ± 0.78	70.99 ± 1.90	38.69 ±1.02	19.55 ± 1.23
PLs4	97.64 ± 3.05	55.02 ± 1.08	37.28 ±0.87	17.46 ± 0.64
PLs5	97.64 ± 0.95	62.08 ± 0.62	36.21 ±0.84	15.63 ± 0.18
PLs6	99.69 ± 0.87	60.95 ± 1.85	32.13 ±0.78	16.05 ± 0.35
PLs7	66.42 ± 0.18	39.18 ± 0.94	23.28 ± 5.16	16.24 ± 0.14
PLm1	156.24 ± 4.76	90.89 ± 0.07	68.05 ±1.54	32.34 ± 0.71
PLm2	151.32 ± 1.57	91.86 ± 0.08	88.69 ±0.15	50.93 ± 0.12
PFo	23.74 ± 0.10	12.36 ± 0.65	5.49 ±0.63	2.26 ± 1.06
PCh1	24.76± 0.28	10.12 ± 0.55	4.31 ±0.41	Not active
PCh2	25.78 ± 0.15	11.09 ± 0.34	6.89 ± 0.25	4.13 ± 0.98
PPs	34.54 ± 0.38	22.76 ± 0.78	11.21 ±0.72	3.51 ± 0.66
PIk	97.40 ± 3.05	34.66 ± 0.36	18.10 ±0.53	10.29 ± 0.64
POi	14.17± 0.33	4.57 ± 0.64	0.84 ±0.87	Not active
PSa	62.07 ± 1.05	40.02 ± 0.75	21.68 ± 0.65	11.74 ± 0.86

Values expressed are means ± S.D. of three parallel measurements. GAE: gallic acid equivalent.

**Table 4 molecules-27-08198-t004:** Collection areas of the NEAR samples.

Sample Code	Collection Area
PLs1-PLs7	Lesvos
PLm1-PLm2	Lemnos
PCh1-PCh2	Chios
PFo	Fourni
PPs	Psara
PIk	Ikaria
POi	Oinousses
PSa	Samos

## Data Availability

Not applicable.

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
