# Peer review of "Assessment of Antioxidant and Antimicrobial Properties of Selected Greek Propolis Samples (North East Aegean Region Islands)"

_molecules, 2022, doi:10.3390/molecules27238198_

Round 1

Reviewer 1 Report

Propolis is natural product collected from plant resins by bees and it is one of the richest sources of polyphenols. The antioxidant effect of the propolis is based on polyphenols and it is closely related to the structure of these compounds. Moreover, it exhibits pluripotent biological activity including antibacterial properties. The content of diterpens influences on the antimicrobial activity. Due to diversity in composition of propolis depending on several factors: botanical sources, bee species, collection season, the conducted study seems especially valuable.

The reviewer suggests major revisions. The list of suggestions and remarks are below:

1.       In part 1 - Results  Authors should add explanation of abbreviations propolis samples  in Table 1. (page 3). The mentioned remark should be introduced into Table 2 ( page 4)

Moreover, in Table 3 - table header should be explain. %DPPH inhibition included in mentioned table should be described. Authors should explain indicated concentrations.

2.       In part 3, Authors should better refer to obtained results and add more studies, so proper literature should be provided.

In lines 153-156 Authors should explain precisely their observation.

In line 172-174 Authors should add concentrations of propolis.

3.       In part 4 - Materials and method - Author missed weight of propolis samples . Moreover, the section 4.7 entitled;” DPPH assay” should be better described. Authors indicated three different propolis extracts concentrations.

4.       Last part – Conclusions should be re-written. I don’t understand the mentality of Authors. The aim of the research was to assess the chemical and biological diversity of propolis, therefore the authors should focus on this aspect.

Author Response

The list of suggestions and remarks are below:

  • In part 1 - Results  Authors should add explanation of abbreviations propolis samples  in Table 1. (page 3). The mentioned remark should be introduced into Table 2 ( page 4)

The abbreviations of propolis samples are explained in table 4 (part 4.1). Moreover, an abbreviation list has been added according to Reviewer 2 suggestion.

  • Moreover, in Table 3 - table header should be explain. %DPPH inhibition included in mentioned table should be described. Authors should explain indicated concentrations.

The explanation for the DPPH assay has been added in part4.7. DPPH (2,2-DiPhenyl-1-PicrylHydrazyl ) assay

The concentrations of propolis samples for the DPPH assay are referred in the Table 3 and there are (200μg/mL, 100μg/mL and 50μg/mL)

The %DPPH inhibition is explained in the part 4.7: “The % inhibition of the DPPH radical for each dilution was calculated using the following formula: %Inhibition = {[1-(A-AB)]/AT}×100, where A is the absorbance of the sample, AT the absorbance of control and AB the absorbance of sample without the DPPH radical.”

  • In part 3, Authors should better refer to obtained results and add more studies, so proper literature should be provided.

In part 3 the obtained results are deeply discussed in comparison with 10 different references. The phytochemical profile is discussed in first and second paragraph and the conclusion regarding the type of propolis has been achieved. Furthermore in the third paragraph the chemical profile is connected with the vegetation of the area from previously published manuscripts. The correlation between the diterpenic profile and the antimicrobial and radical scavenging activity is discussed in paragraph 4.Tthe bioassays are further discussed in the following paragraphs.

  • In lines 153-156 Authors should explain precisely their observation.

It is precisely explained in the Results part2.2 lines100-106 “Moreover, an association was found between the levels of diterpenes of all samples and the antimicrobial activity (Figure 2). Spearman’s rank correlation coefficients were -0.88 to -0.64 indicating a strong correlation between the concentration of diterpenes and antimicrobial activity (P<0.005). In Fig. 2 red and blue colors represent negative and positive Spearman’s rank correlation coefficients between compound concentration levels and activity. In the case of antimicrobial activity, the negative correlation means that the antimicrobial activity is higher when diterpene levels are higher” and in part2.3 lines121-123 “Furthermore, sugars seem to correlate (rs 0.52-0.58) positevely (blue color in Fig. 2) with TPC and DPPH inhibition, while diterpenes were inversely (red color in Fig. 2) correlated as expected (P<0.01).

  • In line 172-174 Authors should add concentrations of propolis.

Concentrations of propolis have been added: “The DPPH radical scavenging activity of the extracts at the concentration of 200 μg/mL showed...”

  • In part 4 - Materials and method - Author missed weight of propolis samples. Moreover, the section 4.7 entitled;” DPPH assay” should be better described. Authors indicated three different propolis extracts concentrations.

In part 4.2  the weight of propolis samples (10 g) was added.

The DPPH assay was studied in three different concentrations (200μg/mL, 100μg/mL and 50μg/mL) as it is described in part 4.7 “different propolis extracts (4, 2, 1 mg/mL) were prepared using DMSO as a solvent. In a 96-well plate 10 μL of each sample were mixed with 190 μL of DDPH solution (12.4 mg / 100 mL in ethanol) and then incubated, at room temperature, for 30 minutes strictly in darkness” Moreover it is explained that:“The % inhibition of the DPPH radical for each dilution was calculated using the following formula: %Inhibition = {[1-(A-AB)]/AT}×100, where A is the absorbance of the sample, AT the absorbance of control and AB the absorbance of sample without the DPPH radical.”

  • Last part – Conclusions should be re-written. I don’t understand the mentality of Authors. The aim of the research was to assess the chemical and biological diversity of propolis, therefore the authors should focus on this aspect.

Conclusions have been revised following the suggestion and trying to overview the scope and perspectives of this study.

Reviewer 2 Report

The manuscript entitled " Assessment of antioxidant and antimicrobial properties of selected Greek propolis samples (Northeast Aegean Region islands)" presented by Pyrgioti et al, summaries a research study towards establishment of antioxidant and antimicrobial activity of propolis sample. Overall, the manuscript is addressing and delivering the scientific content. However, some major flaws need to be addressed for further improvement:   

·       Title can be improved in terms of more meaningful name of propolis or some understandable meaning of propolis…may be “bee based balsamic and resinous substances” in place of (Northeast Aegean Region islands)

·       Please reframe the sentences “It is characterized by an 28 alterable chemical composition as it is contingent on the plant source of each region and 29 the collection period” Page 1(Line 28-29)

·       Please check and reframe the sentences “According to recent literature, propolis consists of more than 30 180 different types of chemicals [3-5] with well-known health properties according to 31 which has been exploited by humans in medicine and cosmetology since Antiquity” Page 1 (Line 30-32)

·       Please check languages thoroughly and reframe more meaningful sentences.

·       May be North East Aegean Region (NEAR) in place of Northeast Aegean Region (NEAR)

·       It is better to provide separate list of abbreviation

·       In material and methods 16 samples are reported, however in table 4 only 9 samples code is mentioned…….please check for uniformity throughout the manuscript.

·       Please check write scientific method for extraction in place of “immersion” (Page 7, Line 186)

·       Please mention the quantity obtained for each compounds after isolation with their chemical structure.

·       Why author chosen total phenolic content method….instead of terpenoids content…as extract was found to be rich with terpenoids.

·       Why antioxidant through only DPPH ?

·       Why only GCMS method not LCMS…as author are working on Ethanol extract, which contain mixture of both polar and non-polar components.

·       Author must check the way of scientific writing by following similar kind of study presented and conducted earlier  

·       Language and any other typological mistake can be address

Author Response

  • Title can be improved in terms of more meaningful name of propolis or some understandable meaning of propolis…may be “bee based balsamic and resinous substances” in place of (Northeast Aegean Region islands)

Propolis is a very famous subject with more than 8500 published papers (scopus) so the term ‘propolis” is trivial and understandable in the title of a paper.

  • Please reframe the sentences “It is characterized by an 28 alterable chemical composition as it is contingent on the plant source of each region and 29 the collection period” Page 1(Line 28-29)

The sentence (Line 28-29) is rephrased “It is characterized by a variable chemical composition as it depends on the plant source of each region and the collection period”

  • Please check and reframe the sentences “According to recent literature, propolis consists of more than 30 180 different types of chemicals [3-5] with well-known health properties according to 31 which has been exploited by humans in medicine and cosmetology since Antiquity” Page 1 (Line 30-32)

The sentence (Line 30-32) is rephrased: “Following the literature, propolis consists of more than 180 different chemical compounds [3-5] and it is well-known for its health properties, according to which propolis has been exploited by humans in medicine and cosmetology since Antiquity [6]”

  • Please check languages thoroughly and reframe more meaningful sentences.

The manuscript has been checked and several rephrases have been introduced.

  • May be North East Aegean Region (NEAR) in place of Northeast Aegean Region (NEAR)

It has been replaced throughout the manuscript accordingly

  • It is better to provide separate list of abbreviation

A list of abbreviations has been added, even though the terms used are very common in bee-keeping products (such as propolis) studies

  • In material and methods 16 samples are reported, however in table 4 only 9 samples code is mentioned.please check for uniformity throughout the manuscript.

In Table 4 all the sixteen samples are referred but some of them are from the same collection area and they are mentioned in one row e.g. PLs1-Pls7, which means that all these seven samples are collected from Lesvos Island.

  • Please check write scientific method for extraction in place of “immersion” (Page 7, Line 186)

The term “maceration” has been used instead

  • Please mention the quantity obtained for each compounds after isolation with their chemical structure.

The quantities of the isolated compounds have been added in part 4.4 and the chemical structures have been added in the supplementary file.

  • Why author chosen total phenolic content method instead of terpenoids content…as extract was found to be rich with terpenoids.

In propolis samples usually a mixture of terpenoids, phenolic acids and flavonoids exists. In this base, the TPC assay is usually used in the bibliography for the evaluation of the phenolic content. The authors used this method in order to have comparable results with previously published data from their team but also with the general bibliography that use this evaluating method.

  • Why antioxidant through only DPPH ?

The design of this study was to get some first principles on antioxidant and antimicrobial activities together with chemical analysis of the NEAR propolis samples. DPPH surely is working-horse towards screening of antioxidant activity and as in our case the antioxidant properties were not extremely strong, we did not try further assays.

 Why only GCMS method not LCMS…as author are working on Ethanol extract, which contain mixture of both polar and non-polar components.

Because of the very complex chemical composition of propolis, GC-MS became the most often used method in the 1980s for rapid chemical characterization of propolis samples of different geographic and plant origins (Greenaway, et al.,Bee World 1990, 71, 107–118.). However, most of the constituents of propolis are relatively polar (flavonoids, phenolic acids and their esters, etc.), and silylation is necessary to increase their volatility and enable GC analysis. This circumstance, combined with the advent in the 1990s of soft ionization techniques compatible with liquid chromatography, soon made HPLC-DAD and HPLC-MS the favorite methods for analysis of propolis phenolic constituents (Hamazaka, et al., Food Sci. Technol. Res. 2004, 10, 86–92.; Ahn et al., Food Chem. 2007, 101,1383–1392; Medana, et al., Phytochem. Anal. 2008, 19, 32–39). Nevertheless, the unprecedented resolving power of capillary GC and the valuable structural information provided by EIMS still provoke scientists to use GC-MS despite the disadvantages of derivatization procedures (Popova et al., Phytomedicine 2005, 12, 221–228; Fernandez et al., J. Agric. Food Chem. 2008, 56, 9927–9932; Kalogeropoulos et al., Food Chem. 2009, 116, 452–461; Garcia-Viguera et al., Z. Naturforsch. C 1993, 48, 731–735).

According to very recent overview on propolis among most well accepted analytical techniques for establishing the chemical profiles of propolis samples are TLC and GC–MS (Kasote et al. Nov 2022).

Towards the potential future uses of propolis in the pharmaceutical area, European Pharmacopoeia appreciates a lot the use of GC-MS analyses as easy, reproducible and low cost methods to be followed by scientist and companies instead of other more sophisticated ones. Moreover, the results by GC-MS could be easier compared with numerous existing ones.

  • Author must check the way of scientific writing by following similar kind of study presented and conducted earlier  

With full respect to the Reviewer’s opinion, the authors always follow similar kind of studies among which are the ones by themselves in this scientific field, moreover, a very recent ref (Nov 2022) that has been also added in the References (Kasote et al 2022)

  • Language and any other typological mistake can be address

 The language has been checked thoroughly and amendments have been introduced accordingly

Round 2

Reviewer 1 Report

The manuscript still have required corrections.

Table 3 still requires clarification and correction of the format in such a way that the table does not contain empty cells and unlabeled columns. The first two cells in the first column should be merged and there should be information about the type of data presented in this column (eg: Samples). The first two cells in the second column should also be merged. In addition, a description should be added for columns 3, 4 and 5 (under the heading "% DPPH inhibition") as it is not clear which substance the concentrations of 200, 100 and 50 µg / mL apply to. It should also be clarified in chapter 4.6. "Total phenolic content" why in Table 3 the concentrations in columns 3, 4 and 5 are 200, 100 and 50 µg / mL, respectively.

Author Response

Dear reviewer thank you for your suggestions regarding the format.

In Table 3 the first two cells in the first column have been merged and the information “Samples” has been added.

The first two cells in the second column have been merged.

Table 3 shows the results from two different tests for the same propolis samples. The results from TPC, which are expressed as mg Equivalent to Gallic Acid (GAE) per gram of dry extract (method is explained in the part 4.6) and the DPPH inhibition, which is expressed as % inhibition of the DPPH radical for each dilution (part 4.7).

The propolis samples (as they are mentioned in the first column) are tested in three different concentrations (200, 100 and 50 µg / mL) for the DPPH assay. These concentrations are the final concentration of propolis samples in the 96-well plate. The method “different propolis extracts (4, 2, 1 mg/mL) were prepared using DMSO as a solvent. In a 96-well plate 10 μL of each sample were mixed with 190 μL of DDPH solution …” explains how we prepare these concentrations and it is also mentioned in the following ref [Stavropoulou, et al., NMR Metabolic Profiling of Greek Propolis Samples: Comparative Evaluation of Their Phytochemical Compositions and Investigation of Their Anti-Ageing and Antioxidant Properties. J. Pharm. Biomed. Anal. 2021, 194, 113814, doi:10.1016/j.jpba.2020.113814.]

Reviewer 2 Report

Manuscript is improved now.

Author Response

Thank you